# In Vivo Ultrasound Prediction of the Fillet Volume in Senegalese Sole (*Solea senegalensis*)

**DOI:** 10.3390/ani12182357

**Published:** 2022-09-09

**Authors:** João Afonso, Cristina Guedes, Alfredo Teixeira, Paulo Rema, Severiano Silva

**Affiliations:** 1Faculdade de Medicina Veterinária, ULisboa, Avenida da Universidade Técnica, 1300-477 Lisboa, Portugal; 2Veterinary and Animal Research Centre (CECAV) and Associate Laboratory of Animal and Veterinary Science (AL4AnimalS), University of Trás-os-Montes e Alto Douro, Quinta de Prados, 5000-801 Vila Real, Portugal; 3Centro de Investigação de Montanha (CIMO), Instituto Politécnico de Bragança, Campus de Santa Apolónia, 5300-253 Bragança, Portugal; 4Laboratório Associado para a Sustentabilidade e Tecnologia em Regiões de Montanha (SusTEC), Instituto Politécnico de Bragança, Campus de Santa Apolónia, 5300-253 Bragança, Portugal; 5CIIMAR—Interdisciplinary Centre of Marine and Environmental Research, University of Porto, Novo Edifício do Terminal de Cruzeiros de Leixões, Avenida General Norton de Matos, s/n, 4450-208 Matosinhos, Portugal

**Keywords:** fillet volume, fillet yield, ultrasound, *Solea senegalensis*

## Abstract

**Simple Summary:**

The ability to obtain in vivo information on characteristics related to fish composition is necessary for aquaculture. In addition, there is growing interest in production traits, such as growth, feed efficiency or fillet weight, but it remains difficult to precisely record in vivo individual fish traits that report to these production traits, which can increase edible fish meat production and decrease the environmental impact. In the present study, we performed an ultrasound approach for the in vivo prediction of fillet volume of the Senegalese sole (*Solea senegalensis*), a species considered a promising flatfish species for marine fish farming. The results show that models based on ultrasound fillet volume measurements explain above 95% of the variation observed in fillet volume. However, for fillet yield estimation, the results were modest. Therefore, further studies are necessary to better understand the potential of the ultrasound approach to this trait. Nevertheless, this work allows us to conclude that the approach with ultrasound is promising for measuring in vivo fish composition traits.

**Abstract:**

Senegalese sole (*Solea senegalensis*) has been considered a promising new flatfish species for Mediterranean marine fish farming. Accurate prediction of fillet traits in live animals may allow for more efficient control of muscle deposition in fish. In this sense, this study was undertaken to develop a non-invasive method to predict in vivo fish fillet volume and yield using real-time ultrasonography (RTU). The trial was conducted with 44 market weight Senegalese sole (298.54 ± 87.30 g). Fish were scanned with an Aloka SSD 500V with a 7.5 MHz probe. Ten RTU cross-sectional images were taken from the operculum to the caudal fin at regular intervals. These images were analyzed using Fiji software. These data were then used to estimate the partial volumes of the fillet. Actual fillet volume was determined using Archimedes’ principle. Simple and stepwise multiple regression analyses were then used to develop prediction models of fillet volume and yield. The most cranial RTU sections of the fish fillet were the best single predictors of both fillet volume and fillet yield and were the ones included in the best stepwise models. The best RTU slice area explained 82% of the variation observed in fillet volume, but the other RTU slice areas used as predictors of fillet volume showed poor to moderate accuracy (0.035 ≤ R^2^ ≤ 0.615). Single RTU partial volumes showed poor to very high accuracy (0.395 ≤ R^2^ ≤ 0.970) as predictors of fillet volume. The best stepwise model based on the RTU slice areas included three independent variables and explained 88.3% of the observed variation. The best stepwise models based on RTU partial volumes (single volumes and/or combinations of single volumes) explained about 97% of the variation observed in fillet volume. Two RTU volume traits, V_1–5_ + V_6–9_, and V_1+()+9_, showed to be practically direct predictors of the actual fillet volume, explaining, respectively, 97% and 96% of the variation observed in the actual fillet volume. The fillet yields show lower correlations with slice areas (r between 0.044 and 0.601) than with volumes (r between 0.288 and 0.637). While further studies are clearly necessary to better understand the potential of RTU for the estimation of fillet yield in fish in general and Senegalese sole in particular, the present results showed that RTU traits can be very good predictors of Senegalese sole’s fillet volume, either used in regression models or as direct predictors.

## 1. Introduction

In the EU, there is an orientation to turn the aquaculture sector more competitive by focusing on quality, health and safety and providing a high-quality, highly nutritional and trustworthy product to the consumer in an eco-friendly way [1]. Senegalese sole (*Solea senegalensis*) is one of the species that has deserved attention and has been considered a promising flatfish species for Mediterranean marine fish farming [2]. For fish species, increasing fillet yield will result in more retail products per unit weight of live fish produced. This requires knowledge of the body composition traits in live fish, which are essential for feeding and genetics programs [3,4]. However, such programs depend critically on accurate and, above all, non-invasive methods to predict body composition in live fish [5,6]. For fish, comprehensive studies have been conducted using computed tomography (CT) [7,8,9] or magnetic resonance imaging (MRI) [3,10] to predict fish composition traits. However, despite the indisputable merit of these imaging techniques for estimating the composition of fish and other farm species [11,12], their cost and low portability limit their practical use [13]. Another imaging technique that has been widely used to predict the in vivo composition of meat species is real-time ultrasonography (RTU) [11]. This technique has shown enormous evolution in the last decade and is currently presented as indispensable in meat production as a non-invasive, versatile, simple-to-operate, and low-cost tool to assess meat traits [14]. In addition to the main meat species, this technique has also been successfully used to estimate fish composition traits [15,16,17,18,19]. These works had different objectives and used the RTU to obtain in vivo measurements of muscle and fat in fish. For example, Silva et al. [16] used the RTU and image analysis to assess the fat of Senegalese Sole, while other works used this technique to obtain muscle measurements, such as area [12] or muscle thickness measurements [4,20], with the objective of predicting slaughter yields and predictors for genetic breeding programs. In these works, there were several species, such as Common carp (*Cyprinus carpio* and *C. rubrofuscus*) [4], European sea bass (*Dicentrarchus labrax*) [20], rainbow trout (*Oncorhynchus mykiss*) [5] and tambaqui (*Colossoma macropomum*) [15]. In all the works that used probes from 6 to 10 MHz, it was concluded that the RTU is an effective tool to obtain in vivo body and carcass traits that can be used in feed and breeding programs in a simple, efficient and sustainable way, compared to previously used methods, namely the destructive approaches. Although ultrasound has been used for evaluating animal composition for over 60 years, little information is available about volume measurements obtained in vivo by RTU. Thus, this study was undertaken to predict the fillet volume and fillet yield of Senegalese sole individuals from RTU measurements obtained in vivo using a probe of 7.5 MHz and image analysis.

## 2. Material and Methods

### 2.1. Fish and Experimental Procedures

Forty-four market weight Senegalese sole (*Solea senegalensis*); mean body weight 298.54 ± 87.30 g; range 178.77 to 456.32 g) were used. The fish were chosen randomly from the aquaculture research facilities of the University of Trás-os-Montes and Alto Douro (Vila Real, Portugal). All animals were kept under the same management conditions, stocked in a rectangular PVC tank (bottom area 0.75 m^2^; water column 50 cm; volume: 500 L; water-flow rate: 5.5 L min^−1^), supplied with recirculated seawater (18.7 ± 1.6 °C; salinity: 35.2 ± 1‰), and were fed ad libitum an extruded commercial diet (2 mm) for marine fish supplied by AQUASOJA. Animal handling followed the guidelines of EU directive number 2010/63/EU and FELASA category B and C recommendations concerning the protection of animals used for scientific purposes.

### 2.2. Ultrasound Procedure

Fish were scanned with an Aloka SSD 500V real-time scanner (Aloka, Tokyo, Japan) equipped with a linear array transducer of 7.5 MHz (Aloka UST-5512U-7.5, Tokyo, Japan) with a gain, near gain and far gain of 64, 24 and 0.9, respectively. Figure 1 shows an example of an RTU image. The RTU images were taken in cross-sectional slices (S_1_ to S_10_; Figure 2) from the end of the operculum to the beginning of the caudal fin. To achieve this, the probes were placed perpendicular to the fish major axis and displaced along the fish in a craniocaudal movement from the end of the operculum to the beginning of the caudal fin. To ensure that the transducer was always placed in the same relative position, the length of the fish (mean length 203.11 ± 36.52 mm; range 162.30 to 299.60 mm) was considered. To ensure optimal acoustic contact, a gel standoff was placed between the probe and the fish. Fish were placed in a rectangular metallic container with saltwater (column: 3 cm), and RTU scans were taken under anesthesia using 150 mg L^−1^ phenoxyethanol (Merck–Schucherd, Hohenbrunn, Germany).

### 2.3. Image Acquisition, Analysis and Volume Calculation

The fish length, from the end of the operculum to the beginning of the caudal fin, was determined using a ruler and this length was divided by 10 to obtain the length of each section between two consecutive locations for RTU cross-sectional scanning. As the probe moved along the 10 cross-sectional positions (S_1_ to S_10_; Figure 2), the RTU images for each position were captured. These images were identified and recorded in a TIFF file for further analysis. For each cross-sectional image, the right dorsal fillet area (A_1_ to A_10_) was determined using Fiji software (ImageJ 1.49u) [21] by tracing the contour of the muscle and counting the number of pixels within that contour. The RTU fillet volumes were then calculated using the Cavalieri principle [22] procedure (see Figure 2)—V_1_, V_2_, V_3_, V_4_, V_5_, V_6_, V_7_, V_8_ and V_9_, for sections between each pair of consecutive RTU slices; V_1–5_, for a section between slices S_1_ and S_6_; V_6–9_, for a section between slices S_6_ and S_10_; V_1–3_, for a section between slices S_1_ and S_4_; V_4–6_, for a section between slices S_4_ and S_7_; V_7–9_, for a section between slices S_7_ and S_10_. Briefly, the volume of each section (RTU partial volume) was calculated multiplying the mean fillet areas of each pair of RTU images by the section length, and for volumes between non-adjacent slices, only the areas of the limiting slices of that section are used (e.g., for V_1__–__5_, only are used area measurements from S1 and S5). The RTU fillet volume (V_1+()+9_; cm^3^) was calculated using the following equation:(1)$$Volume=\overset{9}{\underset{i=1}{\sum }}A_{i}d_{i}\;\;\;$$
where *A* is the mean section area (cm^2^), *d* is the section length (cm) and *i* is the number of sections.

### 2.4. Carcass Dissection and Fillet Volume Determination

After ultrasound images were recorded, the fish were euthanized with an overdose of 500 mg L^−1^ phenoxyethanol (Merck–Schucherd, Hohenbrunn, Germany). Body weight (BW) and body length measurements were recorded. The fish volume was also determined using Archimedes’ principle. Then, the right dorsal fillet of each fish was dissected. This dissection was meticulous enough to have a standardized fillet extract procedure. To minimize fillet shape deformation, fillets were placed over a horizontal surface. The fillet was weighed and fillet yield was calculated as the ratio between fillet weight and fish weight. The fillet volume was determined according to Archimedes’ principle—the fillets were submerged in water and the volume of water displaced by this action was measured.

**Figure 2 animals-12-02357-f002:**
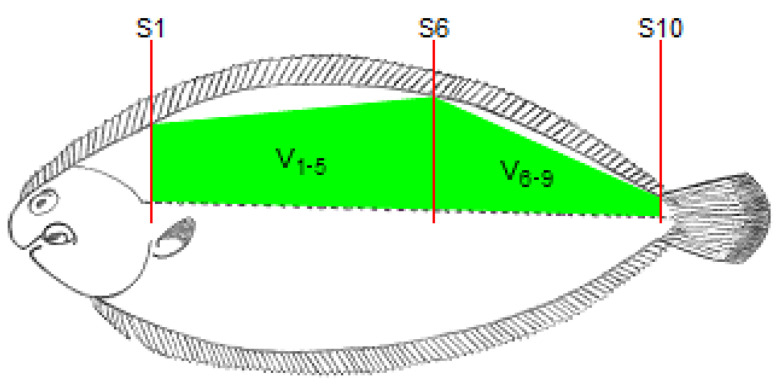
Schematic representation of the location of the cross-sectional RTU slices (S_1_, S_2_, S_3_, S_4_, S_5_, S_6_, S_7_, S_8_, S_9_ and S_10_).

### 2.5. Statistical Analysis

Descriptive statistics, correlation and simple and multiple regression analysis were performed with JMP-SAS software (v.14; JMP-SAS Institute Inc. Cary, NC, USA). Multiple stepwise forward regression with k-fold cross-validation models were developed to predict fillet volume (dependent variable) from RTU partial volumes (independent variables). A 10-fold cross-validation technique was used to assess the stability of the regression models. The accuracy of the estimates was based on the coefficient of determination (R^2^), and the root mean square error of the cross-validation (RMSE) was used to determine the precision of the prediction models. The best models were chosen based on a high R^2^ and lowest RMSE.

## 3. Results 

### 3.1. Descriptive Statistics 

Fish body weight (BW) presented a range of variation of 178.77 to 456.32 g (Table 1). Fillet weight and fillet volume exhibited larger CV (35.0% and 35.6%, respectively) than those of BW and total volume (29.2% and 28.4%, respectively). Fillet yield showed a much smaller CV (9.6%). As expected, given the body shape of the Senegalese sole, the most cranial RTU sections are those with the largest dimensions. Concerning all RTU traits estimated, the ones with the CV closest to that of fillet volume were V_5_ and V_1–5_ (CV = 35.2%), V_1+()+9_ (CV = 35.1%) and V_1–3+4–6+7–9_ (CV = 35.9). Overall, the CV for single RTU partial volumes (29.1% ≤ CV ≤ 51.2%) was larger than the CV for slice areas (18.2% ≤ CV ≤ 39.4%) but quite close to the CV for combinations of RTU partial volumes (29.0% ≤ CV ≤ 45.2%).

### 3.2. Prediction of Fillet Volume from RTU Area(s) and Volume(s)

Fillet volume estimates based on A_2_ showed high accuracy (R^2^ = 0.820; *p* < 0.01; Table 2), but the other estimates based on RTU slice areas showed very poor to moderate accuracy (0.035 ≤ R^2^ ≤ 0.615; *p* < 0.01; Table 2), while the estimates of fillet volume based on single RTU partial volumes showed poor to very high accuracy (0.395 ≤ R^2^ ≤ 0.970; *p* < 0.01; Table 2). For corresponding locations, single RTU partial volumes showed significantly higher accuracy than RTU slice areas. In addition, the precision of fillet volume estimates based on single RTU partial volumes was higher than that based on RTU slice areas (3.08 ≤ RMSE ≤ 13.79 and 7.51 ≤ RMSE ≤ 17.40, respectively; *p* < 0.01; Table 2). Both types of estimates (based on RTU slice areas or on single RTU partial volumes) showed a trend toward more accurate and precise results when obtained from the cranial part of the fish body. This was particularly clear considering V_1–5_ (R^2^ = 0.970; RMSE = 3.08; *p* < 0.01; Table 2) and V_6–9_ (R^2^ = 0.802; RMSE = 7.88; *p* < 0.01; Table 2), and V_1+2+3_ (R^2^ = 0.956; RMSE = 3.73; *p* < 0.01; Table 2), V_4+5+6_ (R^2^ = 0.912; RMSE = 5.24; *p* < 0.01; and V_7+8+9_ (R^2^ = 0.679; RMSE = 10.05; *p* < 0.01). Confirming this trend, the combinations of RTU partial volumes based only on the last sections of the fish body (V_7+8_ and V_7+8+9_) showed moderate accuracy as predictors of the fillet volume (respectively R^2^ = 0.626 and R^2^ = 0.679; *p* < 0.01; Table 2), against the high to very high accuracy shown by all the other combinations of RTU partial volumes (0.853 ≤ R^2^ ≤ 0.960; *p* < 0.01; Table 2). 

### 3.3. Prediction of Fillet Volume Using Stepwise Multiple Linear Regression 

The stepwise analysis confirmed that the trend for RTU data obtained from the cranial part of the fish body was the best predictor of fillet volume (Table 3). All stepwise models explained a very large amount of the variation observed in the fillet volume. Still, also in line with the results from simple linear regression analysis, the best stepwise model based on RTU slice areas explained a significantly less amount of variation (R^2^ = 0.8830; *p* < 0.01; Table 3) than any stepwise model based on RTU partial volumes (0.9689 ≤ R^2^ ≤ 0.9755; *p* < 0.01; Table 3). As indicated by the simple linear regression analysis, V_1–5_ was the best RTU predictor of fillet volume, and the stepwise analysis showed that the inclusion of other RTU partial volumes, with or without V_1–5_ in the final model, at best, provided a slight improvement in the amount of variation observed. As indicated by the simple linear regression analysis, V_1–5_ was the best single RTU predictor of fillet volume, and the stepwise analysis showed that, using different combinations of the several RTU predictors analyzed, in the final model, at best, only provided a slight improvement in the amount of variation observed. Together with the stepwise results for models with combinations of RTU volumes, this shows that the location of the RTU measurements used is more important than the amount of information used to obtain the final fillet volume estimates. 

Concerning V_1–5_ (33.39 ± 11.75 cm^3^) and V_6–9_ (14.52 ± 5.47 cm^3^), it must be pointed out that the sum of the mean RTU fillet volumes obtained with these two traits was equal to the mean RTU fillet volume (V_1+()+9_; 47.91 ± 16.81 cm^3^) and such a value wasn’t significantly different from the actual mean fillet volume (49.12 ± 17.51 cm^3^), with V_1–5_ + V_6–9_ and V_1+()+9_ explaining, respectively, 97% (*p* < 0.01; data not shown and 96% (*p* < 0.01; Table 2) of the variation observed in the actual fillet volume.

### 3.4. Correlations between Fillet Yields and RTU Slice Area(s) and Volume(s)

The correlations between the fillet yields and the RTU traits are presented in Table 4. The correlation values found for the fillet yield and the fillet yield_V1–5_ are similar, although 19 out of 32 correlations studied were higher for the latter yield. In general, correlations based on RTU volumes tended to show higher correlations (r between 0.288, *p* > 0.05 and 0.637, *p* < 0.01 for fillet yield and r between 0.346, *p* < 0.05 and 0.609, *p* < 0.01 for fillet yield_V1–5_) than those based on RTU slice areas (r between 0.055, *p* > 0.05 and 0.601, *p* < 0.01 for fillet yield and r between 0.044, *p* > 0.05 and 0.408, *p* < 0.01 for fillet yield_V1–5_). Furthermore, the highest correction values were observed with measure V_1_ (r = 0.637, *p* < 0.01) and with measure V_1+2_ (r = 0.637, *p* < 0.01) for fillet yield and fillet yield_V1–5_, respectively. These results show the importance of the measurements obtained in cranial positions. 

## 4. Discussion

The larger RTU slice areas (and consequently larger RTU volumes) obtained in the cranial part of the fish body reflect the larger volume of the fillet there, given the body shape of the Senegalese sole. Therefore, such RTU data are expected to be better predictors of total fillet volume, as observed. According to this suggestion, depending on the fish shape (and muscle volume/mass distribution along the body), the single RTU slice areas shown to be the best predictors of fillet volume or weight will vary for different fish species. This is in line, for instance, with the results of Bosworth et al. [23], with catfish pointing at an RTU slice location about midway the tip of the head and the beginning of the tail fin as the best RTU predictor of fillet weight, as could be expected since catfish have a much more regular fusiform shape than Senegalese sole. The shape and size of the animals are essential for accuracy when using the RTU technique. In the present work, the size of the fish was adequate for the size of the probe, and in this way, it was possible to obtain images that allowed the area of the fillet section to be well framed. This aspect was discussed for other species as a factor that compromises the accuracy of the prediction models [14,24]. In these works, several solutions are pointed out to overcome the problem of probe length, such as the superposition of images using reference points in the images or the use of lengthier probes but generally of lower frequency, which compromises the ability to analyze the most superficial tissues. The best single-variable regression model developed by Bosworth et al. [23] for shank fillet in females showed considerably less accuracy (R^2^ = 0.50) than the present best single RTU slice area model. This can be related to the smaller number of ultrasound slice locations tested by Bosworth et al. [23]. As could be expected from the single regression analysis, Bosworth et al. [23] best three-variable regression model also showed smaller accuracy (R^2^ = 0.56) than the present best RTU slice areas stepwise model. In both cases, there were three independent variables, but the slice locations selected were more evenly distributed across the fish body in Bosworth et al. [23] study, reflecting the more regular fusiform shape of catfish. 

The higher accuracy and precision of single RTU partial volumes as predictors of fillet volume, when compared with RTU slice areas, can be partially explained by the larger amount of the RTU information used (two RTU slice areas for each RTU partial volume). However, the use of the distance between each pair of RTU slice areas per single RTU partial volume, which is related to the length of the fillet, also contributed to this result.

Gonçalves et al. [17] had already obtained very good estimates of fillet volume of Tilapia, (R^2^ = 0.89) but inferior to the best ones now obtained for Senegalese sole. This comparison is particularly relevant concerning the estimates now obtained with V_1+()+9_ or with V_1–5_ + V_6–9_, since in these two cases, as in Gonçalves et al.’s [17] study, the RTU measurements used covered the entire length of the fish fillet. The difference observed in the amount of variation explained can be due to differences in the shape of the fish body (and, consequently, in the shape of the fish fillet) or to a slight difference in the estimation of total RTU fillet volume, since Gonçalves et al. [17] simply multiplied each RTU area by the mean distance between slices to obtain each partial RTU volume, while in the present study, each partial RTU volume was obtained multiplying the average of the two consecutive RTU areas by the distance between them. All the final models obtained by stepwise multiple regression were very good or even excellent predictors of fillet volume. For application in commercial conditions, the model just with V_1–5_, being an excellent predictor of fillet volume, seems to be quite practical. However, such an approach, relying on a prediction equation, requires a relatively homogeneous population in terms of body characteristics, such as body shape, for the model to apply. This means that the use of RTU methodology for use in research studies of genetic variation is limited at the start. This is even more regrettable, given the potential of a technology that allows in vivo collection of data. The present results seem to allow a different approach, without the need for predictive regression models, considering that V_1–5_ + V_6–9_, and V_1+()+9_, correspond practically to direct estimates of the actual fillet volume, in the case of V_1+()+9_, with a correlation of 98% with the fillet volume. The use of volume measurements (3D) in prediction models of carcass and body composition has been valued with different techniques in meat species [25,26,27] and fish [6]. In the latter study, a 3D model of common carp was developed to determine the best morphological predictors of slaughter yields. The accuracy for predicting yields with 3D is superior to or equal to those obtained by 2D predictors, but the practical advantages in obtaining information in the field are pointed out. Nevertheless, and in general, the works that use 3D measurements indicate an advantage over one-or two-dimensional measurements in the prediction models, and it is anticipated that with equipment and software advances, it will be possible to speed up all processes to obtain accurate models based on 3D images to predict in vivo composition traits [28].

The results of the current study are consistent with the findings of several studies that showed ultrasound measurements provide a poor prediction of fillet yield. For example, Flick et al. [29] pointed out fillet yield as an important trait for genetic improvement of fish production efficiency and Sang et al. [30] best model explained 77% of the variation observed in fillet yield for river catfish, with the fish volume adjusted for body weight accounting, just by itself, for 74%. However, Vandeputte et al. [20], using a linear combination of proportions of surfaces and ratios of ultrasound measurements to body length to estimate the same trait in European sea bass, showed poor accuracy (0.02 ≤ R^2^ ≤ 0.18) of fillet yield estimates. So did Maas et al. [9] used linear and CT measurements to estimate fillet yield of mirror carp not gutted before the CT was done (0.08 ≤ R^2^ ≤ 0.45). These poor accuracies are in line with the present results and relate to the fact that most of the variation observed in fillet weight is explained by the variation observed in fish weight. Vandeputte et al. [31] had already pointed out fillet yield being mostly proportional to body weight as one of the main problems in selecting for increased fillet yield. Rutten et al. [32], for instance (whose best model, using length, height, width and corrected length, only explained 15% of the variation observed in fillet yield), showed a correlation of 97% between body weight and fillet weight in Nile tilapia, resulting in a much smaller CV for fillet yield than for fillet weight or volume. As in the present study, Rutten et al. [32], Vandeputte et al. [20] and Maas et al. [9] obtained a much higher CV value for fillet weight (33.3%, 32.8% and 37.1%, respectively) than for fillet yield (7.6%, 2.8% and 5.7%, respectively). Compared to these values, Sang et al. [30] showed a similar CV value for fillet weight (CV = 35.6) and a much higher CV value for fillet yield (CV = 19.7). This much higher CV value for fillet yield shown by Sang et al. [30] may explain why the best model explained a much larger amount of the variation observed in fillet yield, but the reason for such a difference in that CV value, species related or not, remains to be clarified.

## 5. Conclusions

The present study showed that RTU slice areas and RTU partial fillet volumes (single or in different combinations) can be good predictors of Senegalese sole’s total fillet volume and can be used in either single or multitrait regression models. Models based on RTU partial fillet volumes can provide particularly accurate estimates of total fillet volume, either used in regression models or as direct predictors. In the latter case, this makes it possible not only to monitor changes in fillet volume, regardless of eventual changes in body/fillet proportions throughout the growing process of the fish, but also to exploit these eventual changes to achieve genetic progress. In addition, it opens the possibility to apply the same methodology to live animals of different species, breeds or genetic lines, for instance, without the need to develop specific regression models for each one. Concerning fillet yield, the poor results now obtained seem to confirm the results of previous studies showing a close relation between fillet volume/weight and body volume/weight, and further studies are necessary to better understand the potential of RTU for estimating fillet yield in fish in general and Senegalese sole in particular. Still, depending on the heritability values and genetic correlations between fillet volume and fillet yield, the former may prove to be a useful predictor trait for fillet yield, but this clearly needs further research. 

## Figures and Tables

**Figure 1 animals-12-02357-f001:**
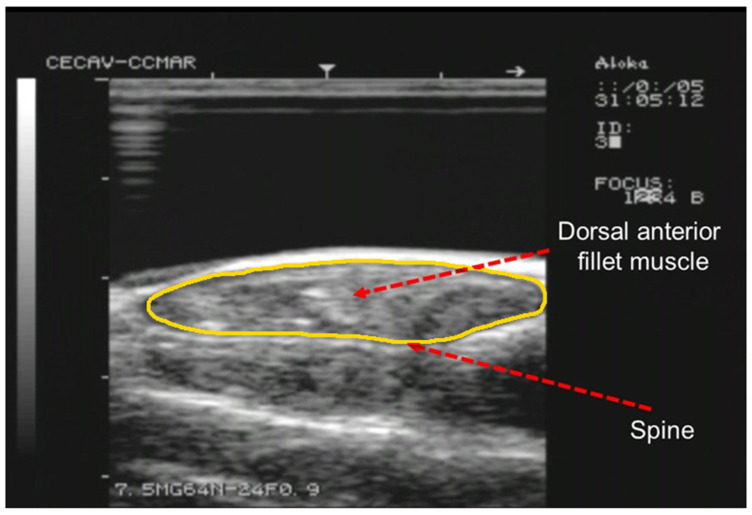
Example of a RTU cross-sectional image acquired with a 7.5 MHz probe at S_3_ position. Arrows indicate the spine and right dorsal fillet muscle. The outline of the fillet section is highlighted in yellow.

**Table 1 animals-12-02357-t001:** Mean, standard deviation (sd), minimum, maximum and coefficient of variation (CV) for fish and real-time ultrasonic (RTU) traits (n = 44).

Trait	Mean	SD	Minimum	Maximum	CV (%)
Fish					
Body weight (g)	298.54	87.30	178.77	456.32	29.2
Total volume (cm^3^)	283.27	80.59	172.10	412.70	28.4
Length (mm)	203.11	36.52	162.30	299.60	18.0
Right dorsal fillet weight (g)	49.31	17.28	26.90	85.00	35.0
Right dorsal fillet volume (cm^3^)	49.12	17.51	28.20	88.20	35.6
Right dorsal fillet yield (%)	16.32	1.57	12.08	20.13	9.6
RTU traits for the right dorsal fillet					
Slice area (cm^2^) *					
A_1_	3.40	0.90	2.15	5.74	26.5
A_2_	3.68	1.15	2.07	6.45	31.3
A_3_	3.60	0.66	1.98	4.68	18.2
A_4_	2.75	0.57	1.51	4.06	20.8
A_5_	2.61	0.56	1.49	3.55	21.6
A_6_	2.62	0.69	0.89	4.13	26.1
A_7_	1.96	0.60	0.95	3.91	30.4
A_8_	1.40	0.43	0.71	3.11	30.5
A_9_	1.02	0.40	0.24	2.25	39.4
A_10_	0.81	0.22	0.48	1.44	27.4
Single volumes (cm^3^) *					
V_1_	7.12	3.11	3.89	15.26	43.7
V_2_	7.80	3.82	3.45	19.32	48.9
V_3_	7.43	2.31	3.25	12.99	31.0
V_4_	5.64	1.64	2.59	8.77	29.1
V_5_	5.41	1.91	2.56	10.10	35.2
V_6_	5.48	2.24	1.52	10.48	40.8
V_7_	4.05	1.69	1.85	9.27	41.9
V_8_	2.86	1.05	1.35	5.91	36.7
V_9_	2.13	1.09	0.41	5.36	51.2
V_1–5_	33.39	11.75	17.31	62.54	35.2
V_6–9_	14.52	5.47	7.33	29.50	37.7
Volume combinations (cm^3^)					
V_1+2_	14.92	6.74	8.43	33.61	45.2
V_3+4_	13.07	3.79	5.87	21.20	29.0
V_5+6_	10.89	3.94	4.08	20.58	36.2
V_7+8_	6.91	2.52	3.75	13.79	36.5
V_1+2+3_	22.35	8.78	11.72	44.75	39.3
V_4+5+6_	16.53	5.31	8.18	28.43	32.1
V_7+8+9_	9.03	3.41	4.74	19.15	37.8
V_1+3+5+7+9_	26.12	8.96	13.85	50.22	34.3
V_2+4+6+8_	21.79	7.98	10.78	41.85	36.6
V_1–3+4–6+7–9_	51.98	18.65	25.18	98.31	35.9
V_1+()+9_	47.91	16.81	24.64	89.04	35.1

* See Figure 2 for location of the RTU slices and volumes in the fish.V_1+2_ = sum of V_1_ and V_2_; V_3+4_ = sum of V_3_ and V_4_; V_5+6_ = sum of V_5_ and V_6_; V_7+8_ = sum of V_7_ and V_8_; V_1+2+3_ = sum of V_1_, V_2_ and V_3_; V_4+5+6_ = sum of V_4_, V_5_ and V_6_; V_7+8+9_ = sum of V_7_, V_8_ and V_9_; V_1+3+5+7+9_ = sum of V_1_, V_3_, V_5_, V_7_ and V_9_; V_2+4+6+8_ = sum of V_2_, V_4_, V_6_, and V_8_; V_1–5_ = volume between slices S_1_ and S_6_; V_6–9_ = volume between slices S_6_ and S_10_; V_1–3_ = volume between slices S_1_ and S4; V_4–6_ = volume between slices S_4_ and S_7_; V_7–9_ = volume between slices S_7_ and S_10_; V_1–3+4–6+7–9_ = sum of V_1–3_, V_4–6_ and V_7–9_; V_1+()+9_ = sum of V_1_, V_2_, V_3_, V_4_, V_5_, V_6_, V_7_, V_8_ and V_9_.

**Table 2 animals-12-02357-t002:** Simple linear regressions between the right dorsal fillet volume and real-time ultrasonic (RTU) traits for the right dorsal fillet (n = 44).

Variables		Intercept	Slope	R^2^	RMSE
Dependent	Independent				
Fillet volume (cm^3^)	RTU slice areas (cm^2^) *				
	A_1_	−2.53	15.20	0.613	11.02
	A_2_	−1.63	13.79	0.820	7.51
	A_3_	−12.50	17.13	0.411	13.60
	A_4_	13.83	12.82	0.175	16.09
	A_5_	2.13	18.03	0.337	14.43
	A_6_	−3.43	20.05	0.615	10.99
	A_7_	20.22	14.76	0.252	15.32
	A_8_	38.33	7.69	0.035	17.40
	A_9_	25.85	22.90	0.274	15.10
	A_10_	21.57	34.18	0.186	15.98
	Single RTU volumes (cm^3^) *				
	V_1_	12.75	5.11	0.826	7.40
	V_2_	15.04	4.37	0.908	5.38
	V_3_	−2.75	6.98	0.845	6.98
	V_4_	−1.43	8.97	0.707	9.58
	V_5_	7.27	7.74	0.710	9.55
	V_6_	9.02	7.31	0.872	6.35
	V_7_	16.13	8.15	0.622	10.89
	V_8_	19.20	10.46	0.395	13.79
	V_9_	23.63	12.00	0.556	11.80
	V_1–5_	0.13	1.47	0.970	3.08
	V_6–9_	7.49	2.87	0.802	7.88
	Combinations of RTU volumes (cm^3^)				
	V_1+2_	11.95	2.49	0.920	5.01
	V_3+4_	−6.66	4.27	0.853	6.79
	V_5+6_	3.83	4.16	0.877	6.21
	V_7+8_	11.20	5.49	0.626	10.83
	V_1+2+3_	5.57	1.95	0.956	3.73
	V_4+5+6_	−2.92	3.15	0.912	5.24
	V_7+8+9_	10.92	4.23	0.679	10.05
	V_1+3+5+7+9_	−0.42	1.90	0.942	4.25
	V_2+4+6+8_	2.59	2.14	0.947	4.06
	V_1–3+4–6+7–9_	3.18	0.88	0.887	5.96
	V_1+()+9_	0.23	1.02	0.960	3.55

* See Figure 2 for location of RTU slices and volumes in the fish. V_1+2_ = sum of V_1_ and V_2_; V_3+4_ = sum of V_3_ and V_4_; V_5+6_ = sum of V_5_ and V_6_; V_7+8_ = sum of V_7_ and V_8_; V_1+2+3_ = sum of V_1_, V_2_ and V_3_; V_4+5+6_ = sum of V_4_, V_5_ and V_6_; V_7+8+9_ = sum of V_7_, V_8_ and V_9_; V_1+3+5+7+9_ = sum of V_1_, V_3_, V_5_, V_7_ and V_9_; V_2+4+6+8_ = sum of V_2_, V_4_, V_6_, and V_8_; V_1–5_ = volume between slices S_1_ and S_6_; V_6–9_ = volume between slices S_6_ and S_10_; V_1–3_ = volume between slices S_1_ and S4; V_4–6_ = volume between slices S_4_ and S_7_; V_7–9_ = volume between slices S_7_ and S_10_; V_1–3+4–6+7–9_ = sum of V_1–3_, V_4–6_ and V_7–9_; V_1+()+9_ = volume between slices S_1_ and S_10_.

**Table 3 animals-12-02357-t003:** Best models for predicting fillet volume, obtained after stepwise multiple regression with k-fold cross-validation, considering area and volume measurements (n = 44) *.

Stepwise Model	Intercept	Independent Variables					R^2^	RMSE
With RTU slice areas	−22.414	7.329 A_1_	7.904 A_2_	6.734 A_5_			0.8830	5.9901
With RTU single volumes	1.519	0.964 V_1_	1.604 V_2_	1.404 V_3_	1.625 V_4_	1.575 V_6_	0.9715	2.9573
With combinations of 2 RTU volumes	0.317	1.331 V_1+2_	1.459 V_3+4_	0.908 V_5+6_			0.9727	2.8934
With combinations of 3 RTU volumes	0.867	1.296 V_1+2+3_	1.167 V_4+5+6_				0.9725	2.9060
With V_1–5_, V_6_, V_7_, V_8_ and V_9_	1.655	1.246 V_1–5_	1.592 V_6_	−1.008 V_8_			0.9755	2.8403
With V_1_, V_2_, V_3_, V_4_, V_5_ and V_6–9_	1.171	1.395 V_1_	1.834 V_2_	1.777 V_3_	1.863 V_4_		0.9653	3.2585
With V_1–5_ and V_6–9_	0.127	1.467 V_1–5_					0.9689	3.0832

* See Figure 2 for location of the RTU slices and volumes, indicated by the subscripts, in the fish. V_1+2_ = sum of V_1_ and V_2_; V_3+4_ = sum of V_3_ and V_4_; V_5+6_ = sum of V_5_ and V_6_; V_1+2+3_ = sum of V_1_, V_2_ and V_3_; V_4+5+6_ = sum of V_4_, V_5_ and V_6_; V_1–5_ = volume between slices S_1_ and S_6_; V_6–9_ = volume between slices S_6_ and S_10._

**Table 4 animals-12-02357-t004:** Correlations between fillet yields and real-time ultrasonic (RTU) traits (n = 44).

Independent *	Dependent	
	Fillet Yield (%) #	Fillet Yield_v1–5_ (%) ##
RTU slice areas (cm^2^)		
A_1_	0.601	0.450
A_2_	0.529	0.480
A_3_	0.253	0.305
A_4_	0.055	0.099
A_5_	0.176	0.308
A_6_	0.345	0.199
A_7_	0.164	0.298
A_8_	0.152	0.044
A_9_	0.187	0.174
A_10_	0.095	0.133
Single RTU volumes (cm^3^)		
V_1_	0.637	0.590
V_2_	0.597	0.595
V_3_	0.495	0.516
V_4_	0.377	0.408
V_5_	0.431	0.514
V_6_	0.517	0.433
V_7_	0.399	0.494
V_8_	0.288	0.346
V_9_	0.381	0.359
V_1–5_	0.582	0.591
V_6–9_	0.466	0.468
Combinations of RTU volumes (cm^3^)		
V_1+2_	0.633	0.609
V_3+4_	0.464	0.491
V_5+6_	0.501	0.494
V_7+8_	0.387	0.475
V_1+2+3_	0.616	0.402
V_4+5+6_	0.489	0.493
V_7+8+9_	0.409	0.466
V_1+3+5+7+9_	0.562	0.584
V_2+4+6+8_	0.546	0.535
V_1–3+4–6+7–9_	0.555	0.561
V_1+()+9_	0.559	0.566

* See Figure 2 for location of the RTU slices and volumes in the fish. # Fillet yield = (Fillet weight/Body weight) × 100; ## Fillet yield_V1–5_ = (Estimate Fillet volume from V_1–5_/Body weight) × 100. Correlation values below 0.298 are not significant (*p* > 0.05); between 0.299 and 0.387 are significant (*p* < 0.05) and above 0.388 are very significant (*p* < 0.01). V_1+2_ = sum of V_1_ and V_2_; V_3+4_ = sum of V_3_ and V_4_; V_5+6_ = sum of V_5_ and V_6_; V_7+8_ = sum of V_7_ and V_8_; V_1+2+3_ = sum of V_1_, V_2_ and V_3_; V_4+5+6_ = sum of V_4_, V_5_ and V_6_; V_7+8+9_ = sum of V_7_, V_8_ and V_9_; V_1+3+5+7+9_ = sum of V_1_, V_3_, V_5_, V_7_ and V_9_; V_2+4+6+8_ = sum of V_2_, V_4_, V_6_, and V_8_; V_1–5_ = volume between slices S_1_ and S_6_; V_6–9_ = volume between slices S_6_ and S_10_; V_1–3_ = volume between slices S_1_ and S4; V_4–6_ = volume between slices S_4_ and S_7_; V_7–9_ = volume between slices S_7_ and S_10_; V_1–3+4–6+7-9_ = sum of V_1–3_, V_4–6_ and V_7–9_; V_1+()+9_ = volume between slices S_1_ and S_10_.

## Data Availability

Not applicable, as this study did not report any data.

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
