# Peer review of "In Vivo Ultrasound Prediction of the Fillet Volume in Senegalese Sole (Solea senegalensis)"

_animals, 2022, doi:10.3390/ani12182357_

Round 1

Reviewer 1 Report

The manuscript by Afonso et al. presents a study using real-time ultrasonography to predict the fillet volume and fillet yield of Senegalese sole (Solea senegalensis). A total of 44 market-weight fish were anaesthetized, and 10 cross-sectional transdermal ultrasound images were captured using a 7.5MHz linear transducer. Image analysis was then performed to calculate the cross-sectional area of the dorsal fillet for each of the 10 images, and fillet volume was estimated by applying the Cavalieri principle using cross-sectional area values. This is a small study (N = 44), but the study design, data acquisition and data analysis seem to have been conducted in a sound and solid way, and the approach and results should be of interest to the readers of ‘Animals’. However, the manuscript is quite brief in parts, would benefit from including some more detail in several areas, and could be more clearly structured . Please see below for specific comments.

Abstract:

The abstract is clear and concise and adequately summarises the study design and results.

Introduction:

Concise manuscripts are excellent however the introduction is too brief, and fails to give a sufficient overview of the topic, and therefore fails to place the present study in adequate context. For example, the authors state that “Although ultrasound has been used for evaluating animal composition for over 60 years, there is little information available about volume measurements obtained in vivo by RTU”. However, in the discussion, the authors refer to studies (e.g. Bothworth et al. and Sang et al.) which DO predict fillet traits in live fish. I therefore suggest that the introduction be re-written to provide a better summary of the literature and justification for why this study is unique and relevant.

Materials and methods:

Line 75: States that the study performed on twenty-four (24) fish, but elsewhere in the manuscript states forty-four (44).

Line 86: what was the penetration depth of the ultrasound? And what were the gain settings used?

Line 87: Figure 1: Suggest including the outline of the fillet area as determined by Image J. From the picture, it looks like the entire cross-section of the fillet cannot be captured by the ultrasound probe? Will this have implications for the accuracy of predictions? And does this vary with fish size (i.e. are you unable to capture the entire cross-sectional area of the fillet in large fish compared to small fish?) If this is the case, this should be addressed in the discussion.

Line 93: Remove results from the M&M. This should be presented in the results section.

Line 96: Details of the manufacturer of phenoxyethanol should be included.

Line 99: Table 1 should be placed in the Results section not the M&M. I suggest that figures 1&2 should be placed in the M&M before Table 1.

Line 117: Details of the software developer (National Institute of Health) should be included.

Line 121: When calculating partial volumes between non-adjacent slices e.g. V1-5, are only the area measurements from S1 and S5 used, or all areas between S1 and S5 used? i.e. Is V1-5 calculated as (A1 + A5)/2 x distance between S1 and S5 (as described in Figure 2), or is it V1 + V2 + V3 + V4 + V5 which is how I interpret the text? If it is the former, and only 2 ultrasound images are needed (at S1 and S2) this is an easy way of estimating fillet volume quickly and easily. Suggest making this distinction clearer in the text.

Line 132: How was fillet yield calculated?

Line 133: “Total volume” is recorded in Table 1. Was the volume of the whole fish determined using the Archimedes principle as well? If so, this should be included in the M&M.

Line 143: More information about the stepwise regression should be included. Is this forwards or backwards stepwise regression? And how was the “best” model chosen? Was this the model with the best R2, the lowest root mean square error, both?

Line 145: Remove “obtained with a 7.5 MHz probe” as this has already been introduced.

Line 149: Why have the authors used RSDcv as the abbreviation for root mean square error? RSD is normally used for relative standard deviation (SD/mean). Maybe RMSE or would be more appropriate?

Results:

The results of this study are very interesting. However, I find the text in this section a little confusing with so many figures and subscripts, and I had to read it several times to properly understand it. Can the text be simplified to really highlight the results that the authors think are most important? There is also quite a lot of duplication of results between the tables and the text. Perhaps adding some subheadings to the results sections may help clarify the results? For example:

3.1 Prediction of fillet volume from RTU slice area(s)

3.2 Prediction of fillet volume from RTU volume(s)

3.3 Prediction of fillet volume using stepwise multiple linear regression

3.4 Prediction of fillet yield from RTU slice area(s) and volume(s)

The authors mention in the M&M that correlations were calculated (Line 142). I think it would be interesting to present the correlations between all results (all areas, partial volumes etc.), perhaps as a correlation matrix? Especially as the authors make the excellent point in the discussion (Lines 266-269) that prediction models have limitations, and that direct correlations are far more practical. A comparison of the results of correlations regression models would also be interesting.

Line 160: How were P-values calculated. This should be described in the M&M.

Table 3: What are the last 3 rows in the table? These data do not have a description in column 1 equivalent to the first 4 rows. Why have these been included? Are they the best combinations of single volumes and partial volumes? This should be included in the text of column 1. Also, what does the “e” mean? Is this an error term?

211: Given that body weight was also measured (and is also practical to do in the field on live fish), why did the authors not try to estimate fillet yield by dividing estimated fillet volume (e.g. from V1-5) by body weight? Furthermore, what was the correlation between actual fillet volume and fillet yield? If this is low, it follows that estimated fillet volume will be a poor predictor of fillet yield. Perhaps it would be useful to also include correlations between actual fish traits as well as ultrasound-predicted traits (see previous comment)?

Discussion:

The discussion is interesting and concise, but like the introduction is a bit brief.

Line 244: While partial volume is a better predictor of fillet yield than area, does this need to be balanced against the speed and ease of collecting and analysing the data? A2 explained over 80% of the variation in fillet volume and has a relatively low RMSE – could it be that this is “good enough” given that only a single ultrasound image is needed? Perhaps this should be discussed?

Line 264: This is only a small study from a single population. Do the authors think the results require external validation on an independent population? Especially as the authors mention the limitations of relying on prediction models.

267-269: This ‘direct comparison data’ (another way of saying correlation?) isn’t presented as only regression model results are shown.

Line 270: Genetic improvement is mentioned several times throughout the manuscript, however no genetic parameters are referenced. Is there any data available for the heritability of fillet volume compared with fillet yield in aquaculture species? And what are the genetic correlations between fillet volume and fillet yield? While the phenotypic correlations may be relatively low, if the genetic correlations are high enough then estimated fillet volume may be a useful predictor trait for fillet yield.

A bit more context and discussion about the potential applications of the results of this study would also be useful. If producers and breeders are more interested in fillet yield than fillet volume, and fillet yield can be predicted relatively well from fish volume adjusted for fish weight, why are we worried about ultrasound? If, however, fillet volume is considered a useful trait, this should be discussed.

Line 279: See earlier comment about calculating fillet yield from estimated fillet volume divided by fillet weight. Why try to predict yield from area or volume when these are known not to be strongly correlated?

Lines 270-293: This paragraph is interesting, but a lot of it reads more like a literature review/introduction rather than a discussion of the results of this study. Perhaps consider moving some of the background information about relationships between fillet weight/yield/volume into the introduction, and reframe this paragraph to have more of a focus on the results of the current study? For example, Lines 270-277 do not reference results of the current study at all. Instead, the authors may consider starting the paragraph by focussing on how their results relate to the literature: “The results of the current study are consistent with the findings of several studies that showed ultrasound measurements provide a poor prediction of fillet yield. For example…”

Conclusion: Clear and concise.

Line 299: Again the authors reference “direct predictors”, but only regression model results are reported in the manuscript. Direct correlation results should be included if they are going to be discussed.

Author Response

Comments and Suggestions for Authors

The manuscript by Afonso et al. presents a study using real-time ultrasonography to predict the fillet volume and fillet yield of Senegalese sole (Solea senegalensis). A total of 44 market-weight fish were anaesthetized, and 10 cross-sectional transdermal ultrasound images were captured using a 7.5MHz linear transducer. Image analysis was then performed to calculate the cross-sectional area of the dorsal fillet for each of the 10 images, and fillet volume was estimated by applying the Cavalieri principle using cross-sectional area values. This is a small study (N = 44), but the study design, data acquisition and data analysis seem to have been conducted in a sound and solid way, and the approach and results should be of interest to the readers of ‘Animals’. However, the manuscript is quite brief in parts, would benefit from including some more detail in several areas, and could be more clearly structured . Please see below for specific comments.

Answer: The authors are grateful for the article's positive comments; with the changes introduced, we are confident that the article has improved.

Abstract:

The abstract is clear and concise and adequately summarises the study design and results.

Answer: Thank you for this very positive appraisal.

Introduction:

Concise manuscripts are excellent however the introduction is too brief, and fails to give a sufficient overview of the topic, and therefore fails to place the present study in adequate context. For example, the authors state that “Although ultrasound has been used for evaluating animal composition for over 60 years, there is little information available about volume measurements obtained in vivo by RTU”. However, in the discussion, the authors refer to studies (e.g. Bothworth et al. and Sang et al.) which DO predict fillet traits in live fish. I therefore suggest that the introduction be re-written to provide a better summary of the literature and justification for why this study is unique and relevant.

Answer: New text was introduced with information that shows the importance of using RTU as a non-invasive technique to measure body and carcass traits.

Materials and methods:

Line 75: States that the study performed on twenty-four (24) fish, but elsewhere in the manuscript states forty-four (44).

Answer: > Already corrected. There was a typing mistake in line 75. We thank the reviewer.

Line 86: what was the penetration depth of the ultrasound? And what were the gain settings used?

Answer: Information about the probe and the image settings has been introduced. An Aloka model UST-5512U-7.5 probe was used. This probe, as it is of higher frequency, allows for better resolution of the most superficial structures. Furthermore, this probe allows penetration of 4 cm, which is suitable for works like the one presented here, where the relevant information for analysis is from 2 to 3 cm.

Line 87: Figure 1: Suggest including the outline of the fillet area as determined by Image J. From the picture, it looks like the entire cross-section of the fillet cannot be captured by the ultrasound probe? Will this have implications for the accuracy of predictions? And does this vary with fish size (i.e. are you unable to capture the entire cross-sectional area of the fillet in large fish compared to small fish?) If this is the case, this should be addressed in the discussion.

Answer: Thank you for this comment about changing the figure. An outline has been introduced in the figure highlighting the measured fillet area, and the fish are of a size that allows the area of interest to be well framed. This is a significant issue when analyzing area measurements with ultrasound. A new text is included in the discussion.

Line 93: Remove results from the M&M. This should be presented in the results section.

Answer: The table was moved.

Line 96: Details of the manufacturer of phenoxyethanol should be included.

Answer: The manufacturer was included.

Line 99: Table 1 should be placed in the Results section not the M&M. I suggest that figures 1&2 should be placed in the M&M before Table 1.

Answer: Already corrected in sequence of correction about line 93. We thank the reviewer for the suggestion – it really is logical and clearer for the reader. 

Line 117: Details of the software developer (National Institute of Health) should be included.

Answer: Already corrected, with the corresponding change in the “References”.

Line 121: When calculating partial volumes between non-adjacent slices e.g. V1-5, are only the area measurements from S1 and S5 used, or all areas between S1 and S5 used? i.e. Is V1-5 calculated as (A1 + A5)/2 x distance between S1 and S5 (as described in Figure 2), or is it V1 + V2 + V3 + V4 + V5 which is how I interpret the text? If it is the former, and only 2 ultrasound images are needed (at S1 and S2) this is an easy way of estimating fillet volume quickly and easily. Suggest making this distinction clearer in the text.

Answer: The assumption is correct – it is the former situation. Maybe it is clear if, instead of “Briefly, the volume of each section (RTU partial volume) was calculated multiplying the mean fillet areas of each pair of RTU images by the section length.” It is “Briefly, the volume of each section (RTU partial volume) was calculated multiplying the mean fillet areas of each pair of RTU images by the section length and, for volumes between non-adjacent slices, only the areas of the limiting slices of that section are used (e.g. for V1-5, only are used area measurements from S1 and S5).”?

Line 132: How was fillet yield calculated?

Answer: A new text was introduced. The fillet yield was calculated as the ratio between fillet weight and fish weight

Line 133: “Total volume” is recorded in Table 1. Was the volume of the whole fish determined using the Archimedes principle as well? If so, this should be included in the M&M.

Answer: A new text was introduced.

Line 143: More information about the stepwise regression should be included. Is this forwards or backwards stepwise regression? And how was the “best” model chosen? Was this the model with the best R2, the lowest root mean square error, both?

Answer: The best models were chosen based on high R2 and lowest RMSE.

Line 145: Remove “obtained with a 7.5 MHz probe” as this has already been introduced.

Answer: It is removed. We thank the reviewer for the remark.

Line 149: Why have the authors used RSDcv as the abbreviation for root mean square error? RSD is normally used for relative standard deviation (SD/mean). Maybe RMSE or would be more appropriate?

Answer: We agree with the reviewer. It was change throughout the text.

Results:

The results of this study are very interesting. However, I find the text in this section a little confusing with so many figures and subscripts, and I had to read it several times to properly understand it. Can the text be simplified to really highlight the results that the authors think are most important? There is also quite a lot of duplication of results between the tables and the text. Perhaps adding some subheadings to the results sections may help clarify the results? For example:

3.1 Prediction of fillet volume from RTU slice area(s)

3.2 Prediction of fillet volume from RTU volume(s)

3.3 Prediction of fillet volume using stepwise multiple linear regression

3.4 Prediction of fillet yield from RTU slice area(s) and volume(s)

Answer: The authors are grateful for the suggestion that was considered with the introduction of the subheadings pointed out in the results sections may help clarify the results.

The authors mention in the M&M that correlations were calculated (Line 142). I think it would be interesting to present the correlations between all results (all areas, partial volumes etc.), perhaps as a correlation matrix? Especially as the authors make the excellent point in the discussion (Lines 266-269) that prediction models have limitations, and that direct correlations are far more practical. A comparison of the results of correlations regression models would also be interesting.

Answer: We thank you for the reviewer's comment. We agree with the reviewer that a correlation analysis is also a very suitable approach for this type of work. However, we opted for a simple regression analysis, which allowed us to understand how RTU measurements explain the variation of fillet volume and, as the correlation allows us to understand the strength of the relationship between two numerical variables.

Line 160: How were P-values calculated. This should be described in the M&M.

Answer: The Software package that calculates regressions returns the p-values of the regressions.

Table 3: What are the last 3 rows in the table? These data do not have a description in column 1 equivalent to the first 4 rows. Why have these been included? Are they the best combinations of single volumes and partial volumes? This should be included in the text of column 1. Also, what does the “e” mean? Is this an error term?

Answer: We hope it is clearer now. The “e” was a typographical error.

211: Given that body weight was also measured (and is also practical to do in the field on live fish), why did the authors not try to estimate fillet yield by dividing estimated fillet volume (e.g. from V1-5) by body weight? Furthermore, what was the correlation between actual fillet volume and fillet yield? If this is low, it follows that estimated fillet volume will be a poor predictor of fillet yield. Perhaps it would be useful to also include correlations between actual fish traits as well as ultrasound-predicted traits (see previous comment)?

Answer: The authors are grateful for the suggestion. We did the suggested analysis and changed Table 4.

Discussion:

The discussion is interesting and concise, but like the introduction is a bit brief.

Line 244: While partial volume is a better predictor of fillet yield than area, does this need to be balanced against the speed and ease of collecting and analysing the data? A2 explained over 80% of the variation in fillet volume and has a relatively low RMSE – could it be that this is “good enough” given that only a single ultrasound image is needed? Perhaps this should be discussed?

Answer: The practicality of the RTU is important, and new texts highlight this aspect.

Line 264: This is only a small study from a single population. Do the authors think the results require external validation on an independent population? Especially as the authors mention the limitations of relying on prediction models.

Answer: This issue is very relevant for the possibility of applying the RTU technique to fish production regardless of species, and external validation will be considered in other studies to determine the prediction model's reproducibility and generalizability to new and different individuals.

267-269: This ‘direct comparison data’ (another way of saying correlation?) isn’t presented as only regression model results are shown.

Answer: The point that is intended to be made with this 'direct comparison data' is that there seems to be the possibility of avoiding the use of regression equations. It is the same as when CT data was used with the Cavalieri principle in body composition, and the question becomes – was dissection data more reliable than CT data with all the losses and added small errors of carcass dissection? Namely, V1-5 + V6-9, and V1+()+9 are just different way of measuring the actual fillet volume – instead of using the Archimedes principle, we used RTU data. Which one is more accurate? Regression equations developed for one population only will apply to a different population if the correlation between the different independent variables is the same. In the case of V1-5 + V6-9, and V1+()+9  we used "direct" individual RTU measurements without assuming anything of the kind. Although we decided to present regression analysis instead of correlation values, maybe the point we want to stress, trying to attend to the reviewer's good point, can be somehow addressed by adding, as example, the value of the correlation between V1+()+9, with and the fillet volume. Please check the change made in the lines in question.

Line 270: Genetic improvement is mentioned several times throughout the manuscript, however no genetic parameters are referenced. Is there any data available for the heritability of fillet volume compared with fillet yield in aquaculture species? And what are the genetic correlations between fillet volume and fillet yield? While the phenotypic correlations may be relatively low, if the genetic correlations are high enough then estimated fillet volume may be a useful predictor trait for fillet yield.

Answer: The goal of the present paper was not to make a genetic study, but just to analyze a new methodology for estimation of fillet volume and/or fillet yield. The point is very good and it is, indeed, one that deserves further research, given some very good results now shown for fillet volume, versus the poor results shown for fillet yield. Maybe a small note in the conclusions could point out this possibility? Please check the change made in the conclusions.

A bit more context and discussion about the potential applications of the results of this study would also be useful. If producers and breeders are more interested in fillet yield than fillet volume, and fillet yield can be predicted relatively well from fish volume adjusted for fish weight, why are we worried about ultrasound? If, however, fillet volume is considered a useful trait, this should be discussed.

Answer: Volume measurements have been used in several works to estimate carcass and body composition traits. While models based on one or two dimensions are useful for a simple view, 3D based models are valuable because they include much broader information about the carcass or tissues. A text was included in the discussion that highlights the importance of using volume measurements.

Line 279: See earlier comment about calculating fillet yield from estimated fillet volume divided by fillet weight. Why try to predict yield from area or volume when these are known not to be strongly correlated?

Answer:

Lines 270-293: This paragraph is interesting, but a lot of it reads more like a literature review/introduction rather than a discussion of the results of this study. Perhaps consider moving some of the background information about relationships between fillet weight/yield/volume into the introduction, and reframe this paragraph to have more of a focus on the results of the current study? For example, Lines 270-277 do not reference results of the current study at all. Instead, the authors may consider starting the paragraph by focussing on how their results relate to the literature: “The results of the current study are consistent with the findings of several studies that showed ultrasound measurements provide a poor prediction of fillet yield. For example…”

Answer: The authors are grateful for the suggestion. The text has been included.

Conclusion: Clear and concise.

Line 299: Again the authors reference “direct predictors”, but only regression model results are reported in the manuscript. Direct correlation results should be included if they are going to be discussed.

Answer: This issue was considered, and correlations were introduced.

Reviewer 2 Report

This manuscript was undertaken to predict the fillet volume and fillet yield of Senegalese sole individuals from RTU measurements obtained in vivo, using a probe of 7.5 MHz, and image analysis. The manuscript is generally well written, and I recommend publication with major revisions. However, there are several scientific concerns about this manuscript.

1. Line 30, “The trial was conducted with 44 market weight Senegalese sole”; Line 75, “Twenty four market weight Senegalese sole were used”. They were contradictory, why?

2. Was the temperature of the PVC tank constant? How long did the whole experiment take, and the temperature have a big effect on the result?

3. The length range of the fish was from 162.30 to 200.60 mm in line 93, but, the maximum value of fish length in Table was 299.60mm, which one was incorrect?

4. Combinations of RTU volumes represent the sum of single RTU volumes, then what is the meaning and purpose of its calculation in the present study?

5. Line208, “the actual actual mean fillet volume” should be replaced by “the actual mean fillet volume”.

6. Lines 194-197, “V1-5 was the best RTU predictor of fillet volume, and the stepwise analysis showed that the inclusion of other RTU partial volumes, with or without V1-5 in the final model, at the best, just provided a slight improvement in the amount of variation observed.” Was there a contradiction between these two results, and why? So which result was reliable? This was not explained in the discussion section.

Author Response

Comments and Suggestions for Authors

This manuscript was undertaken to predict the fillet volume and fillet yield of Senegalese sole individuals from RTU measurements obtained in vivo, using a probe of 7.5 MHz, and image analysis. The manuscript is generally well written, and I recommend publication with major revisions. However, there are several scientific concerns about this manuscript.

  1. Line 30, “The trial was conducted with 44 market weight Senegalese sole”; Line 75, “Twenty four market weight Senegalese sole were used”. They were contradictory, why?

Answer: Already corrected. There was a typing mistake in line 75. We thank the reviewer.

  1. Was the temperature of the PVC tank constant? How long did the whole experiment take, and the temperature have a big effect on the result?

Answer: The temperature conditions of PVC tanks have no implication for the use of the RTU technique in fish. After the fish are placed on the board, capturing the RTU images takes about two minutes.

  1. The length range of the fish was from 162.30 to 200.60 mm in line 93, but, the maximum value of fish length in Table was 299.60mm, which one was incorrect?

Answer: Thank you for the attentive remark. It is a mistake. It was corrected.

  1. Combinations of RTU volumes represent the sum of single RTU volumes, then what is the meaning and purpose of its calculation in the present study?

Answer: Together with the stepwise results for models with combinations of RTU volumes, this shows that it is more important the location of the RTU measurements used than the amount of information used to get the final fillet volume estimates. 

  1. Line208, “the actual actual mean fillet volume” should be replaced by “the actual mean fillet volume”.

Answer: Already corrected. We thank the reviewer.

  1. Lines 194-197, “V1-5was the best RTU predictor of fillet volume, and the stepwise analysis showed that the inclusion of other RTU partial volumes, with or without V1-5in the final model, at the best, just provided a slight improvement in the amount of variation observed.” Was there a contradiction between these two results, and why? So which result was reliable? This was not explained in the discussion section.

Answer: Maybe the text wasn´t very clear. Please check the changes made to see if it is clearer now. Again we thank the reviewer for helping to improve the paper.

Reviewer 3 Report

Research seems interesting and valuable. There are some little comments that can improve the manuscript.

Please check the first affiliation for spelling (also the lines 88 i 96)

Latin names of species should be italic formated (line 26, 56 and others).

Lack of citation for the Cavalieri principle procedure calculation explanation (line 119).

The completeness of the explanation of the abbreviations used in the table require checking.

Would there be a difference in the measurements taken if the fish were euthanizedbefore all measurements?

Author Response

Comments and Suggestions for Authors

Research seems interesting and valuable. There are some little comments that can improve the manuscript.
Answer: Thank you for this very positive evaluation.

Please check the first affiliation for spelling (also the lines 88 i 96)

Answer: It was checked and corrected. We thank the reviewer.

Latin names of species should be italic formated (line 26, 56 and others).

Answer: Already corrected. We thank the reviewer.

Lack of citation for the Cavalieri principle procedure calculation explanation (line 119).

Answer: A reference was introduced.

The completeness of the explanation of the abbreviations used in the table require checking.
Answer: Corrections made in Tables 2 and 4. We thank the reviewer.

Would there be a difference in the measurements taken if the fish were euthanized before all measurements?

Answer: Thank you for the question. From my experience and from what I know of published work, it is not expected that in fish, there will be differences in ultrasound measurements before and after the animals are euthanized. However, in other animal species, such as sheep, there are differences in measurements carried out in vivo and post-mortem, which requires a careful methodology to minimize these differences.

Round 2

Reviewer 2 Report

Through the modification, the article has been improved greatly. This article is recommended for acceptance.